# Prognostic Markers within the Tumour Microenvironment in Classical Hodgkin Lymphoma

**DOI:** 10.3390/cancers15215217

**Published:** 2023-10-30

**Authors:** Arina Martynchyk, Rakin Chowdhury, Eliza A. Hawkes, Colm Keane

**Affiliations:** 1Olivia Newton-John Cancer Research & Wellness Centre, Austin Health, 145 Studley Rd., Heidelberg, VIC 3084, Australia; arina.martynchyk@austin.org.au (A.M.); eliza.hawkes@onjcri.org.au (E.A.H.); 2Princess Alexandra Hospital, 199 Ipswich Rd., Woolloongabba, QLD 4102, Australia; rakin.chowdhury@uqconnect.edu.au; 3Frazer Institute, University of Queensland, St. Lucia, QLD 4072, Australia; 4School of Public Health & Preventive Medicine, Monash University, 553 St Kilda Rd., Melbourne, VIC 3004, Australia

**Keywords:** Classical Hodgkin lymphoma, tumour microenvironment, prognostic markers, PD-L1, immune checkpoint inhibitors, tumour-associated macrophages, T-cells

## Abstract

**Simple Summary:**

Approximately one in seven patients with classical Hodgkin lymphoma have refractory disease or relapse after standard front line chemotherapy. Prognostic biomarkers could help determine candidates for more effective treatment. The latest advances in research techniques have contributed to new insights in understanding the tumour microenvironment of Hodgkin lymphoma and the crucial role it plays in the disease course and response to treatment. Many new potential biomarkers are being explored in this setting and the results of these studies, as well as diagnostic methodologies, are presented in this review.

**Abstract:**

Classical Hodgkin lymphoma (cHL) accounts for 0.4% of all new cancer cases globally. Despite high cure rates with standard treatment, approximately 15% of patients still experience relapsed or refractory (RR) disease, and many of these eventually die from lymphoma-related causes. Exciting new targeted agents such as anti-PD-1 agents and brentuximab vedotin have changed the therapeutic paradigm beyond chemotherapy and radiotherapy alone. Advances in understanding of the molecular biology are providing insights in the context of novel therapies. The signature histology of cHL requires the presence of scant malignant Hodgkin Reed–Sternberg cells (HRSCs) surrounded by a complex immune-rich tumour microenvironment (TME). The TME cellular composition strongly influences outcomes, yet knowledge of the precise characteristics of TME cells and their interactions with HRSCs is evolving. Novel high-throughput technologies and single-cell sequencing allow deeper analyses of the TME and mechanisms elicited by HRSCs to propagate growth and avoid immune response. In this review, we explore the evolution of knowledge on the prognostic role of immune cells within the TME and provide an up-to-date overview of emerging prognostic data on cHL from new technologies that are starting to unwind the complexity of the cHL TME and provide translational insights into how to improve therapy in the clinic.

## 1. Introduction

Classical Hodgkin lymphoma (cHL) accounts for 0.4% of all new cancer cases and 15–25% of all lymphomas [1]. It affects people of all ages, yet young adults are overrepresented as cHL is the most common cancer in those aged under 25 years. In the vast majority, combination chemotherapy is curative; however, a proportion relapse and ultimately die of their disease. Therapeutic strategies are tailored according to baseline clinical features, predominantly the stage, i.e., early versus advanced [2,3], and interim Positron Emission Tomography (PET)-imaging response assessments during initial chemotherapy (escalation strategies in “PET-positive” cases, or de-escalation in “PET-negative” cases). ABVD (doxorubicin, bleomycin, vinblastine, and dacarbazine) and BEACOPP-escalated (bleomycin, etoposide, doxorubicin, cyclophosphamide, vincristine, procarbazine, prednisolone) regimens both achieve high cure rates but at the risk of significant toxicity, particularly in older patients, and thus, one major research focus is on reducing toxicity [4].

Current prognostication relies almost exclusively on clinical risk factors such as disease bulk and number of sites, elevated Erythrocyte Sedimentation Rate (ESR), age, haematological parameters (e.g., lymphocyte count), and B symptoms [5,6,7,8]. Validated prognostic scores for early and advanced disease such as the German Hodgkin Group Study’s early-stage risk stratification (favourable versus unfavourable), Hasenclever International Prognostic Score, and Advanced-Stage Hodgkin Lymphoma IPI determine treatment strategies and estimate survival outcomes with conventional chemotherapy [5,9,10]. New techniques such as measurements of metabolic tumour volume in cHL using segmentation methods on PET are prognostic for outcomes in most cHL studies [11]. While PET imaging has been incorporated into prognostication tools [12], baseline and interim PET radiomics data do not explain the variations in tumour biology that underpin the differing outcomes.

cHL is characterised by malignant Hodgkin Reed–Sternberg cells (HRSCs) surrounded by a complex immune cell infiltrate in the so-called ‘Tumour Microenvironment’ (TME) [13]. The composition of the TME changes with the histological classification of cHL [14]. The most common cHL subtype is nodular sclerosing cHL, characterised by fibroblast-like cells and background fibrosis. Mixed-cellularity cHL, which has a prominent inflammatory cell infiltrate, is the next most common. Lymphocyte-depleted cHL is rare, with a TME predominantly comprising histocytes with background fibrosis. Lymphocyte-rich cHL (LR-cHL), the rarest subtype, has a TME dominated by histocytes and lymphocytes. Epstein Barr Virus-positive (EBV+) cHL may also have unique changes in the TME and more commonly presents with the mixed-cellularity histological subtype. With significant advances in molecular testing technologies, a more profound understanding of the biology of cHL and composition of this immune infiltrate has followed and deepened the understanding of the roles of such factors as the EBV, macrophages, and T-cells.

This enhanced biological understanding has led to not only better disease characterisation but also the incorporation of drugs targeting immune cells—as part of so-called ‘immune checkpoint therapy’—that are postulated to act on key immune cells within the TME. After forty years of chemotherapy, newer immunotherapeutics targeting anti-PD-1 are now an established approach for treating patients with relapsed or refractory cHL (RR cHL) [15], and are being explored in first-line regimens for treatment-naïve disease [16,17,18]. The recent results of the SWOG S1826 study [16] showed the significant benefit of progression-free survival (PFS) in the Nivolumab-AVD arm compared with brentuximab vedotin-AVD (94% vs. 86%, HR 0.48, 99% CI 0.27–0.87, *p* = 0.0005).

The improved understanding of the TME has also led to the evaluation of the influence of TME composition on prognosis using modern laboratory technologies. Here we discuss the literature assessing the prognostic value of components of the cHL TME.

## 2. The HRSC: Mechanisms of Immune Evasion and Interactions with the TME

The characteristic features of cHL are rare malignant HRSCs surrounded by an immune-rich TME. Immune-cell populations that comprise the TME are diverse and, as mentioned above, change depending on the histological subtype, EBV+ or EBV negativity (EBV−), and patient-specific immune responses [14]. Furthermore, the TME promotes the survival, proliferation, and immune evasion of the malignant HRSCs. Key immune-cell populations in the TME include tumour-associated macrophages (TAMs), CD4-expressing or ‘positive’ (CD4+) and CD8-positive (CD8+) T-cell subsets, B-cells, plasma cells, eosinophils, and mast cells [19,20]. There is complex cross talk between HRSCs and the TME through chemokines and cytokines, which, in turn, affect intracellular pathways and further reinforce proliferative and anti-apoptotic growth signals [19].

HRSCs arise from germinal-centre B-cells that lack immunoglobulin gene transcription ability and B-cell programming, leading to a survival advantage and the evasion of apoptosis [21]. The whole-exome sequencing of flow-sorted HRSCs has shown complex chromosomal changes and aneuploidy [22]. In EBV- cHL, these result in a high tumour mutational burden closer to that of solid organ malignancies such as melanoma [22,23]. Interestingly, the presence of a high tumour mutational burden is a predictive factor for immune checkpoint inhibitor (ICI) response in non-haematopoietic malignancies [24].

Genomic changes in HRSCs result in the amplification of cellular pathways that increase cell proliferation. One of the key genomic changes is the upregulation of canonical and non-canonical NF-KB pathways, which results in proliferative and anti-apoptotic cellular signals [25]. Mutations in positive regulators of NF-KB such as the *REL* and *MAP3K14* and the reduced expression of negative regulators and tumour suppressors including *TNFAIP3* lead to the constitutive activation of the NF-KB pathway [22,25]. Interactions between HRSCs and the TME also reinforce intracellular growth signals. For instance, CD30 expressed on HRSCs binds to CD30L on mast cells and eosinophils, which further amplifies intracellular NF-KB pathways [19].

The most frequent chromosomal change in cHL is an arm-level gain of *9p* and a focal amplification of *9p24.1,* leading to the increased expression of the *PD-L1*/*PD-L2* axis [22,26]. Increased programmed death ligand 1 (PD-L1) and programmed death ligand 2 (PD-L2) expression levels on HRSCs constitute a vital mechanism of immune evasion in cHL. These are ligands to the PD-1 receptor and induce T-cell anergy, and so, are critical in shaping the immune synapse between T-cells and HRSCs. However, the increased expression of PD-L1 and PD-L2 is not restricted to HRSCs alone and these ligands are found in other key immune-cell populations within the TME such as TAMs and dendritic cells [27].

A diverse range of interleukins, cytokines, and chemokines are produced by HRSCs, and these recruit immune cells into the TME. Examples of important cytokines and chemokines include IL-5 and CCL28, which recruit eosinophils to the TME and CCL17 (also known as TARC), and CCL22, which attracts CCR4-expressing cells such as type 2 T helper cells and T-regulatory cell subsets (Treg) [19,25]. HRSCs also secrete IL-10 and TGF-beta, which suppresses the activity of cytotoxic T-cells [25]. Furthermore, HRSCs possess receptors for a number of interleukins (such as IL-6), and these act in an autocrine fashion to further reinforce proliferative and survival signals [25,28].

JAK/STAT signalling pathways are amplified in HRSCs, which influences the intracellular responses to cytokine signalling [25]. The focal amplification at *9p24.1* also contains the JAK2 locus, leading to increased *JAK2* expression [22]. Furthermore, there is a constitutive activation of JAK/STAT signalling pathways due to the inactivation of mutations in negative regulators such as *PTPN1* and the activation of mutations of *STAT6* [25,29]. Importantly, JAK/STAT signalling pathways induce additional the increased gene expression of both *PD-L1* and *PD-L2* in HRSCs [30]. Changes in the *PI3K/AKT* pathway are also common in HRSCs (13–40% of cases), and include mutations in *ITPKB* that result in the constitutive activation and enhancement of this pathway [25].

In addition to PD-L1 and PD-L2 overexpression, HRSCs have several adaptations that result in the inhibition of an immune-mediated anti-tumour response. In a study of 108 diagnostic cHL biopsies assessed using immunohistochemistry (IHC), reduced or absent MHC Class I expression was found in 79% (n = 85) of biopsies whilst MHC Class II expression was reduced or absent in 67% (n = 72) of cases [31]. Impaired major histocompatibility complex Class I (MHC-I) expression is common in HRSCs due to the inactivation of mutations in *beta-2-microglobulin* [22]. The absence of or reduced MHC-I expression is a mechanism of immune evasion because CD8+ T-cells require intact MHC-I expression to induce cytotoxic T-cell killing [32]. Major histocompatibility complex Class II (MHC-II) expression is downregulated, but in a smaller subset of cHL, due to rearrangements that result in the loss of function of the *CIITA* gene [22]. At relapse, HRSCs may show the loss of CD58 (11%, n = 6), which normally binds to CD2 and promotes immune adhesion with both NK cells and T-cells [25]. Furthermore, HRSCs express CD95-ligand, which can cause the apoptosis of CD95+ cytotoxic T-cells. The expression of HLA-G and HLA-E is absent in HRSCs, which further inhibits the action of NK cells [25]. These changes in the HRS have profound effects on the TME and remodel the immune synapse resulting in T-cell anergy and the evasion of a tumour-specific response.

## 3. EBV and Changes in the TME in cHL

EBV is present in 30–50% of cHL cases [33]. EBV+ cHL is more prevalent at the extremes of age, in patients with HIV, and in those from developed nations. One study investigated the prevalence patterns of EBV+ cHL patients from the United Kingdom. This showed a bimodal pattern of prevalence, with EBV present in cHL tumour samples in 80% (n = 4) of patients at less than 9 years of age, 24% (n = 37) in patients between 15 to 24 years, and 64% (n = 42) of patients at greater than 49 years of age [34]. The presence of clonal EBV DNA in HRSCs strongly argues for a role of the virus in contributing to the pathogenesis of cHL [35]. EBV+ cHL has a type II latency pattern of antigen expression including LMP1, LMP2, and EBNA1 peptides. This contrasts with post-transplant lymphoproliferative disorders, where a type III latency pattern of antigen expression predominates that has a more extensive expression of viral antigens [36]. LMP1, LMP2, and EBNA1 viral peptides have been implicated in the pathogenesis of EBV+ cHL [35]. For instance, LMP1 functions as a constitutively activated CD40 homolog that amplifies NF-KB, JAK/STAT, AP1, and PI3K/AKT pathways [37]. The importance of these viral-mediated changes to HRSCs is emphasised by the finding that mutations in *TNFAIP3* (negative regulator of NF-KB) are almost mutually exclusive to EBV- HRSCs [22]. EBV+ cHL has significantly fewer driver mutations and a lower mutational burden compared to EBV- cHL [22]. This is likely because the presence of EBV-related viral changes to HRSCs results in less dependence on non-viral cellular changes overall [22].

EBV+ cHL may have unique changes in the TME related to an altered immune response to EBV-related viral peptides. One study using immunohistochemistry showed both increased MHC-I expression in EBV+ cHL compared with EBV- cHL as well as an increased number of activated cytotoxic CD8+ T-cells relative to CD4+ T-cells [38]. The activated CD8+ T-cells were not in the close vicinity of HRSCs, and so, were felt by the authors to be unlikely to exhibit any tumour-specific responses. Another study utilizing flow cytometry also showed increased proportions of CD8+ T-cells and NK cells in EBV+ cHL compared with EBV- cHL, but these findings were not replicated in later studies using single-cell sequencing or gene expression analysis [39,40,41]. The disparate findings may reflect the differences in the analysis of immune cells using flow cytometry compared with single-cell RNA sequencing and gene expression profiling (GEP).

The EBV-related peptide, EBNA1, may promote the recruitment of immunosuppressive FoxP3+ Treg cells into the TME. The GEP of EBV+ HRSCs has shown the upregulation of the chemokine CCL20, which causes the migration of CD4+/FoxP3+ Treg cells into the TME [42]. EBV+ cHL cell lines that lacked EBNA1 expression did not have increased CCL20 expression, but cell lines that retained EBNA1, including cases of EBV+ nasopharyngeal carcinoma, had increased CCL20 expression. These findings suggest that EBNA1 may play an intrinsic role in shaping the TME in EBV+ cHL by increasing CCL20 expression to favour the recruitment of CD4+/FoxP3+ Treg cells that inhibit a tumour-specific immune response.

The association between EBV and clinical outcomes in cHL depends on patient-specific and immune-related factors. A recent meta-analysis showed that EBV+ cHL was associated with worse OS and disease-specific survival (DSS), but there were no significant differences in failure-free survival or event-free survival (EFS) [43]. Subgroup analysis revealed that age influences outcomes in EBV+ cHL; patients aged <15 years have a trend towards better outcomes while those aged >45–50 years have a comparatively poorer OS. In this meta-analysis, there was no statistical difference between outcomes from patients in different geographical locations.

The risk of developing EBV-related malignancies including cHL may also depend on the host’s immune response to EBV infection. The HLA haplotype HLA-A*01 is associated with an increased risk of developing EBV-associated cHL whereas HLA-A*02 is associated with a reduced risk [35]. The cause of this disparate susceptibility to EBV+ cHL may be a difference in the immune response to latent EBV antigens associated with HLA haplotypes. Jones et al. [44] demonstrated that the magnitude of CD8+T-cell response to the latent EBV antigen LMP1/2A was elevated in patients who were HLA-A*02-positive compared to HLA-A*02-negative patients with cHL. The in vitro expansion of LMP2A-specific CD8+ T-cells in healthy volunteers was also largely restricted to those that were heterozygotes for the HLA-A*02 haplotype. This suggests that the reduced risk of EBV+ cHL may be related to a latent antigen-specific CD8+ T-cell response in those who are HLA-A*02-positive.

## 4. Mononuclear Phagocyte System and Dendritic Cells and Their Role in the TME

The mononuclear phagocyte system (MPS) forms the core component of the innate immune system and comprises monocytes, macrophages, and dendritic cells (DCs). The MPS drives several mechanisms underlying both chronic inflammation and healing as well as antigen presentation [45].

### 4.1. Macrophages

Macrophages are essential cells in the innate immune system and can fight against different hazards including cancer cells. There are two main types of macrophages: classically activated macrophages (M1), with pro-inflammatory function, and alternatively activated macrophages (M2), with anti-inflammation activity. M2 macrophages contribute to the suppression of the immune response resulting in cancer progression and are generally identified by the enhanced expression of CD163 [46,47]. Early studies investigating macrophages in the cHL TME utilised IHC with antibodies targeting CD68 and CD163 to delineate M1 and M2 TAMs. A number of these IHC-based studies have assessed the association between TAMs and outcomes [48,49,50,51].

More recent studies based on the integration of several high-resolution techniques including single-cell RNA sequencing, spatial transcriptomics, and multiplex immunofluorescence have confirmed and expanded the previously described data that the MPS in cHL comprises multiple subsets of monocytes, TAMs, and DCs. These subsets are found in specific niches of the TME, and their exact locations define their prognostic value [32,52,53]. Insight into the MPS’ prognostic role has changed dramatically over the past decade due to significant advancements in laboratory techniques.

Macrophages are associated with poor outcomes, but the literature is not consistent. In one of the first analyses, Steidl et al. found an association between CD68+ TAMs in the cHL TME and primary treatment failure and poorer progression-free survival (PFS) [48]. This was the first of a series of studies using different techniques to analyse the prognostic value of TAMs. Another large analysis (n = 288) demonstrated that high CD68 and CD163 expression did correlate with poorer OS and event-free survival (EFS) [49]. Later studies also reported an association between shorter OS and CD163+ TAMs in cHL [54,55]. Contrasting this, Azambuja et al. [56] found no association between tumour IHC expression of CD68 and CD163 and survival.

This contradiction could be explained both by the existence of different types of macrophages as well as by the lower sensitivity of IHC staining to discriminate M1 from M2 macrophages [57]. The polarization of macrophages is much more complex than the M1-and-M2 binary classification and depends on many factors. A shift from M1 to M2 and vice versa, or to a hybrid of both cells, is also possible under certain circumstances [58,59].

The expression on MPS cells of two key immune checkpoints, PD-L1 and PD-L2, has been under scrutiny in cHL given the successful therapeutic targeting of their ligand PD-1 by approved immunotherapies such as nivolumab and pembrolizumab [60,61]. As previously mentioned, cHL HRSCs commonly express these ligands due to the hallmark abnormalities in the *9p24.1 PD-L1/PD-L2* gene responsible for PD-L1/PD-L2 expression [62]. However, within the TME, the highest expression of PD-L1 is in TAMs [63,64]. The spatial relationship between PD-L1+ TAMs and PD-L1+ HRSCs is highly organised. Carey et al. utilised digital spatial profiling (DSP) and multiplex immunohistochemistry to show that PD-L1+ TAMs are in close vicinity to PD-L1+ HRSCs. Strikingly, the mean distance from PD-L1+ TAMs to the nearest PD-L1+ HRSCs was significantly less than the mean distance from PD-L1− TAMs to the nearest PD-L1+ HRSCs in all 20 cases analysed [32]. In addition, Hollander et al. demonstrated that a high proportion of IHC-detected PD-L1+ leukocytes (defined as ≥5%) is independently associated with a statistically significant inferior OS (HR 3.46; 95% CI 1.15–10.37, *p* = 0.03). The authors speculated that PD-L1+ in these leucocytes could be attributed partly to the expression of PD-L1 by macrophages [65].

The expression of the immunomodulatory enzyme indoleamine 2,3-dioxygenase (IDO-1) on TME cells and particularly TAMs has been investigated for its prognostic significance in cHL. The TME areas with IDO-1-positive macrophages or dendritic cells vary markedly and can range from less than 0.1% to 83.4% (median: 1.9%). IDO-1 is an intracellular enzyme, responsible for tryptophan catabolism and the initiation of the production of kynurenines (tryptophan degradation products), that contributes to immune tolerance via pathogenic inflammatory processes [66]. It is produced by tumour cells, dendritic cells, or macrophages. Tumour-cell overexpression of IDO-1 is associated with poor prognosis and a metastatic stage in many cancers [67].

Gene expression (Nanostring nCounter) within the TME of *IDO-1* and *PD-L1* was correlated with macrophage markers *CD68* and *CD163* in 88 cHL samples. High *PD-L1* and *IDO-1* gene expression translated to poor freedom from treatment failure (FFTF), and high *IDO-1* gene expression also translated to poorer DSS and OS. In addition, high *CD68* gene expression correlated with inferior OS whereas *CD163* expression did not. The level of gene expression of *CD68* and *CD163* correlated with the level of protein expression of these markers in 130 cHL samples in the multiplex IHC analysis. High proportions of PD-L1- and IDO-1-expressing TAMs in multiplex IHC were associated with poor FFTF, DSS, and OS. The five-year FFTF was worse in patients with high levels of PD-L1+TAMs (59% versus 85%, *p* = 0.002) and IDO-1+TAMs (71% versus 89%, *p* = 0.003) compared to the patients with low levels of these markers on TAMs. IHC PD-L1+ and IDO-1+ on CD68+ macrophages correlated with EBV+ and advanced-stage and non-NS cHL subtypes but not with age, gender, or IPS. Cox regression analyses demonstrated the high IHC expression of PD-L1+CD68+/CD68+, and IDO-1+CD68+/CD68+ cell ratios, age (≥60 years), and stage (IIB-IV) negatively influenced FFTF independently of other factors. A high PD-L1+CD68+/CD68+ cell ratio and advanced age (≥60 years) also had adversely impacted both DSS and OS whereas EBV-positivity and high IPS (4–7) were associated only with poor OS. In multivariate analysis, both PD-L1+ CD68+/CD68+ ratio and IDO-1+ CD68+/CD68+ ratio independently predicted inferior FFTF [68].

As a surrogate of IDO-1 level in the TME, peripheral blood levels of tryptophan (Trp) and kynurenine (Kyn) have also been measured, and a high serum Kyn/Trp ratio appears to be associated with shorter PFS in cHL and correlated with high levels of IDO-1 IHC expression in the TME [69].

### 4.2. Dendritic Cells

DCs are important antigen-presenting cells participating in processing and presenting antigens to naïve T-cells and, apart from other functions, can promote anti-tumour T-cell responses [70]. To evaluate the prognostic role of DC subtypes within the TME is quite challenging because of their low quantities in tissue samples and the lack of distinctive markers for individual DC subsets [71].

Plasmacytoid DCs (pDCs) and conventional DCs (cDCs) are the two main subtypes of this cell population, and they are distinguished morphologically and functionally. Conventional DCs are also known as myeloid DCs (mDCs) and can be found in peripheral blood. Their prognostic significance in cHL has been evaluated in peripheral blood [72], which is beyond the scope of this review.

Early work showed an association between a low number of DCs in tissue biopsy (detected by IHC CD1a expression being <7%) together with high CD68 IHC expression and worse PFS in patients with primary HL [73]. In another analysis, follicular dendritic cells (FDCs) were identified using IHC with the antibody Ki-FDC1P. The absence of FDCs was associated with inferior outcomes [74].

A recent study utilised spatial transcriptomics and single-cell RNA sequencing to examine the MPS and the interactions between dendritic cells and macrophages with other immune cells in the cHL TME. cDCs were in closer proximity to PD-L1-expressing HRSCs whereas pDCs and activated DCs were not in close spatial proximity to HRSCs. The level of PD-L1 and IDO-1 expression in DCs was close to that of monocytes and macrophages. In addition, there were complex interactions between the cells of the MPS that resulted in the recruitment of immunosuppressive CD4+ T helper cells. Monocytes and macrophages may have a particularly important role as ‘signalling hubs’ that mediate these interactions (refer to Figure 1 for a summary of these interactions). It is speculated that the enrichment of the inflammatory cDC2 monocyte–macrophage niche is associated with early relapse following treatment [52].

To summarise the role of MPS cells in the TME, evidence is mounting that immunomodulatory molecules such as PD-L1 and IDO-1, particularly on macrophages and, to a lesser degree, on other myeloid cells, are associated with poor prognosis in cHL. Understanding the spatial relationship within the TME between these MPS cells and the malignant HRSCs is key to improving patient outcomes and finding a promising target for future therapies.

## 5. T-Cell Subsets of Prognostic Significance in cHL TME

T-cells constitute the most abundant cell type in the cHL TME [75]. CD4+ T-cells far outnumber CD8+ T-cells. In a multicentre study of 63 patients with cHL, formalin-fixed paraffin-embedded (FFPE) biopsy samples were evaluated for the proportions of CD4+ and CD8+ T-cells. This showed that CD4+ T-cells accounted for 48% of all lymphocytes in EBV- cHL whereas CD8+ T-cells accounted for 21% of all lymphocytes [38].

Early studies used IHC to characterise the different T-cell populations and their prognostic significance. Alvaro et al. showed that CD8+ T-cells frequently express TIA-1, which is a marker of cytotoxic potential. However, CD8+ T-cells expressing granzyme B, a marker of activated cytotoxic T-cells, were rare in the vicinity of HRSCs. FoxP3+/CD4+ Treg cells varied in proportion amongst the study cohort. A subset (n = 34/177, 19.2%) of the cohort had a higher proportion of FoxP3+ Treg cells relative to other TME cells (>25 cells per field observed). These findings suggest that whilst CD8+ T-cells were present in the cHL TME, these were not spatially near HRSCs and were quiescent. The authors hypothesised that the presence of immunosuppressive FoxP3+ Treg cells in the TME may be related to the inhibition of a tumour-specific cytotoxic T-cell response. Interestingly, despite the supposed ‘pro-tumoural’ effects of FoxP3+ Treg cells, a higher ratio of TIA-1+/CD8+ T-cells to FoxP3+ Treg cells at diagnosis was significantly associated with worse EFS. A later study showed the similar prognostic significance of the ratio of TIA-1+/CD8+ T-cells to FoxP3+ Treg cells in a cohort of RR cHL patients [76], which was similar to the findings in a study on follicular B-cell lymphoma [77]. These findings contrast with studies in non-haematopoietic malignancies, where higher proportions of CD4+ Treg cells were frequently associated with adverse clinical outcomes [78]. This may be related to differences in the role of FoxP3+ Treg cells in the TME in cHL compared to non-haematopoietic malignancies. Furthermore, later studies have shown that FoxP3+ Treg cells constitute only one of many Treg subsets that can be identified in the cHL TME. In addition, the proportion of TAMs increases in the TME when there is also a high number of CD8+ T-cells, and this may also contribute to the adverse prognosis [41]. This has also been verified in a more recent evaluation of immunohistochemistry markers, which showed that an increased FoxP3-to-CD68 ratio was associated with a trend towards favourable OS [79].

Flow cytometry has also been used to quantify T-cell populations in the cHL TME. Alonzo-Alvarez et al. [80] analysed 92 patients uniformly treated with ABVD chemotherapy and described a high CD4:CD8 T-cell ratio (≥5) that was associated with a significantly worse FFTF. These results initially seemed to contradict the earlier studies using IHC; however, the contrasting results were more likely related to different methodologies and the analysis of total CD4+ cells compared with only CD4+/FoxP3+ Treg cells.

Aoki et al. [40] employed single-cell transcriptomics and DSP to further investigate the T-cells in the TME in cHL with nodular sclerosing or mixed-cellularity subtypes. Consistent with prior studies, CD4+ T-cells were significantly more abundant compared to any other cell type. When compared to reactive lymph node samples, the T-cell populations in cHL had significantly higher proportions of three Treg clusters. The highest CD4+/CD5+ Treg cluster also had an increased expression of the immune checkpoints lymphocyte activation gene 3 (LAG3) and cytotoxic T-lymphocyte protein 4 (CTLA4). There was a higher proportion of PD-1+ cells in non-Treg CD4+ clusters rather than CD8+ T-cells. The non-Treg CD4+ clusters consisted of predominantly type 2 T helper cells. EBV+ cHL had a significantly reduced number of T-cells with a type 17 T helper profile but there were no other T-cell TME differences when compared to EBV- cHL cases.

A distinct population of LAG3+/CD4+ Type 1 Treg cells was found in close spatial proximity to HRSCs that had reduced MHC-II expression [40]. In contrast, FoxP3+/CD4+ Treg cells were more likely to be in close proximity to HRSCs that retained MHC-II expression. Loss of MHC-II expression is known to be a negative predictor of ICI response [26], which further increases the interest in LAG3+ Treg cells as potential mediators of ICI resistance. LAG3 is a ligand of MHC-II. When LAG3+ Treg cells are exposed to HRSC lines that express MHC-II expression, there is a downregulation of LAG3 expression, indicating that the absence of MHC-II expression is necessary for the persistence of LAG3+ Treg cells. In vitro studies also showed that IL-6, the only cytokine with higher gene expression in MHC-II-negative HRSCs, could induce CD4+/LAG3+ Treg cells. A high proportion of LAG3+ Treg cells correlated with a trend towards inferior DSS.

Given LAG3+ and PD-1 are expressed in separate immune clusters, LAG3 and PD-1 dual inhibition may show synergistic activity in cHL. A recent Phase I/II trial of the LAG3 inhibitor favezelimab in combination with pembrolizumab in patients with progressive disease after having used pembrolizumab alone has shown promising efficacy (overall response rate of 31%, median progression-free survival of 9 months) [81]. The results of the phase III Keyform-008 trial (clinicaltrials.gov, NCT05508867) of this new ICI combination are eagerly awaited.

A subsequent study by Aoki and colleagues in the rare LR-cHL subtype showed unique differences in T-cell subsets that comprised the TME when compared to the more common nodular sclerosing and mixed-cellularity subtypes of their earlier study. The key finding was the identification of a unique population of activated T follicular-like cells that are CXCL13+ with the frequent co-expression of PD-1, inducible T-cell costimulator (ICOS), and MHC-II, but lacking CXCR5 expression, distinguishing them from conventional T follicular helper cells [82]. When compared to nodular sclerosing and mixed-cellularity subtypes, CXCL13+ T-cells were significantly enriched in the TME in LR-cHL. DSP revealed that CXCL13+ cells were in close proximity and formed rosettes around HRSCs.

Surrounding the CXCL13+ T-cells were CXCR5+/CD20+ B-cells. One of the functions of CXCL13 is being a B-cell attractant, and it is also a ligand of CXCR5. In single-cell sequencing, the CXCR5+ B-cells had a naïve phenotype, indicating that CXCL13+ T-cells attract naïve B-cells to the cHL TME and prevent these from entering the germinal centre. In vitro analysis revealed that TGF-beta, a cytokine secreted by HRSCs, induced the differentiation of CXCL13+ T-cells from naïve CD4+ T-cells. HRSCs that had high expression levels of TGF-beta, measured using IHC, also had significantly higher numbers of surrounding CXCL13+/PD-1+/CD4+ T-cells. These findings suggest that the TME in LR-cHL is highly organised, with rosettes of CXCL13+ T-cells forming around HRSCs and CXCR5+-naïve B-cells surrounding the CXCL13+ T-cells. Crucially, increased numbers of CXCL13+ T-cells in LR-cHL correlated with a significantly shortened PFS.

It is not known whether targeting the CXCL13/CXCR5 axis will be efficacious in LR-cHL and whether CXCL13+ T-cells are simply coincident with altered HRSC biology in LR-cHL. Indeed, Aoki and colleagues demonstrated that HRSCs in LR-cHL had significantly lower numbers of PD-L1 alterations and that higher PD-L1 expression in HRSCs inversely correlated with the number of CXCL13+ T-cells (refer to Figure 2 for a summary of T-cell populations of prognostic significance).

The role of CD4+ Treg cells with an exhausted phenotype has been evaluated in a number of malignancies [83]. A recent work by Veldman et al. investigated the characteristics of CD4+ T-cells surrounding HRSCs, which are known to lack CD26 expression (a marker of T-cell activation) [84,85]. Using the results of bulk RNA sequencing validated with single-cell RNA sequencing and immunohistochemistry, Veldman et al. showed that CD4+CD26- T-cells are located in close proximity to HRSCs and have a predominantly antigen-experienced Treg memory phenotype. In addition, the CD4+CD26- T-cell population has an increased gene expression of thymocyte selection-associated high mobility group box (TOX) and TOX2 transcription factors. These are known to be upregulated in T-cell states with chronic activation and also result in the increased expression of PD-1 and CXCL13. However, the prognostic effects of CD4+CD26- T-cells, the expression of TOX/TOX2, and the role of Treg cells in modifying ICI response in cHL remain to be better elucidated.

## 6. Prognostic Immune Checkpoint Biomarkers in the TME

TME PD-L1 expression has been associated with adverse outcomes in cHL patients who received chemotherapy alone but also serves as a predictive marker for responsiveness to ICI.

### 6.1. Prognostic Value of PD-1/PD-L1 in Patients Treated with Chemotherapy

The increased co-expression of PD-1 and PD-L1, measured using IHC on baseline cHL tumour samples, was predictive of inferior OS and DSS in patients with newly diagnosed cHL treated with ABVD chemotherapy [86].

The increased expression of other immune checkpoint molecules in addition to PD-L1 has been associated with adverse clinical outcomes. Karihtala et al. [41] performed GEP complemented by multiplex IHC to describe the pattern of immune checkpoint molecule expression within the TME in cHL. The increased gene expression of the immune checkpoint molecules *PD-L1*, *IDO-1*, *LAG-3*, *T-cell immunoglobulin*, *and mucin-domain-containing protein 3* (*TIM3)* correlated with inferior OS in multivariate analysis. This same subset of patients had the increased expression of T-cell cytotoxicity-related genes (measured using a cytolytic score) and macrophage-related genes. Overall, these findings suggest that tumours with increased T-cell cytotoxicity-related gene expression were immunologically active, but the co-existing increased immune checkpoint molecule expression levels constituted an adaptation that ultimately resulted in immune evasion. The composition of the TME within the cohort and the expression of immune checkpoint molecules within cell types were highly heterogenous. There were four distinct immune signatures based on the relative proportions of TAMs and T-cells in the TME. Higher age was associated with an increased proportion of TAMs. A non-inflamed T-cell signature on GEP was associated with the overexpression of macrophage-related genes. However, there was no association with OS based on the relative proportions of TAMs and T-cells and there were no changes in the immune clusters within EBV+ cHL patients. Interestingly, one of the immune clusters had both a high proportion of CD8+ T-cells and TAMs, which may explain the adverse prognosis seen in earlier IHC studies of the TME that had high TIA-1 CD8+ T-cell populations. Cytotoxic T-cells were significantly more likely to express PD-1, IDO-1, and TIM-3 compared to CD4+ T-cells. TAMs that were M2-like (CD163+) had significantly lower expression levels of PD-1 and IDO-1. The higher expression of PD-1 in cytotoxic T-cells compared to total CD4+ subsets contrasted with the earlier study by Aoki et al. [82], which employed single-cell sequencing and DSP. Karihtala et al. [41] used multiplex IHC, which reflects protein rather than gene expression to describe the immune checkpoint profiles of T-cells. In addition, Aoki et al. separated CD4+ T-cell populations into those that were Treg and those that were non-Treg whereas Karihtala et al. evaluated CD4+ T-cells as a single subset in their analysis.

The aberrant expression of components of the PD-1 pathway has been postulated as a logical prognostic marker, plus as a potential predictive marker for anti-PD-1 therapies across many tumour types. The PD-L1 genes are located on chromosome 9p24.1 [87]. As mentioned previously, in cHL, there are several mechanisms resulting in PD-L1 overexpression.

Roemer et al. analysed the influence of 9p24.1 alterations and PD-L1 expression in 108 samples of newly diagnosed cHL undergoing chemotherapy with or without radiation therapy. Fluorescence in situ hybridization (FISH) was used to evaluate samples for genomic changes in 9p24.1 whereas conventional IHC methods were used to evaluate protein expression. A modified H-score to standardise IHC grading was generated by multiplying the percentage of T-cells with the positive staining of PD-L1 (0% to 100%) and average intensity of positive staining in HRSCs (1 to 3+). The H-score (histochemical score) is a semi-quantitative standardised assessment of the IHC expression of any marker, calculated via the summation of the percentage of positive cells multiplied by the intensity of their expression (no staining to strongly staining) [88]. Ninety-seven percent had concordant alterations of the *PD-L1* and *PD-L2* loci (polysomy, 5%; copy gain, 56%; amplification, 36%). There was a wide spectrum of *9p24.1* alterations, from low-level polysomy (6% polysomic HRSCs) to near-uniform *9p24.1* amplification (92% amplified HRSCs). The worst outcome was associated with *9p24.1* amplification compared to other alterations (*p* < 0.001). The incidence of *9p24.1* amplification increased with a clinical risk group and was the highest in the group of patients with advanced disease (50% versus 24% in patients with early-stage favourable and 34% in early-stage unfavourable disease; *p* = 0.024). There was a trend toward worse outcomes in patients who had had *9p24.1* amplification although this marker did not remain prognostic in multivariate analyses [62].

### 6.2. Prognostic Immune Biomarkers in the New Era of ICI

The prognostic value of *9p24.1* genetic alterations in RR cHL patients receiving immunotherapy (nivolumab) was investigated in a biomarker substudy of the CheckMate 205 trial [26]. Genetic abnormalities of *9p24.1* were evaluated via FISH in archival tissue using *CD274 (PD-L1*)- or *PDCD1LG2 (PD-L2)*-specific probes and included centromeric control probes. β2M, MHC-I, and MHC-II expression levels on HRSCs were assessed independently by two expert hematopathologists. The frequency and magnitude of *9p24.1* alterations, including polysomy, copy gain, and amplification, were analysed by reviewing stained FFPE tissue sections and the identification of approximately 50 HRSCs in areas with their highest density. Patients with disease progression on therapy were more likely to have lower-level *9p24.1* alterations (*p* = 0.006) and lower PD-L1 IHC expression on HRSCs using the H-score method (*p* = 0.047) compared to nivolumab responders with CR, PR, or SD. Additionally, patients with lower-level *9p24.1* alterations and lower PD-L1 expression on HRSCs had inferior PFS (*p* = 0.026). In the majority of patients (n = 67, 93%), MHC-I expression was either reduced or absent [26]. MHC-II expression was reduced or absent in 44/72 (61%) patients in the study. There was no association between β2M or MHC-I expression on HRSCs and PFS, confirming that anti-PD-1 therapy responses are likely independent of MHC-I-mediated antigen recognition and CD8+ cytotoxic T-cell responses. Ninety-two percent of patients with CR had tumours with membranous MHC-II expression on HRSCs. These findings suggest that ICI may exert its effect through a non-CD8-mediated effect. There was an association between prolonged PFS and positive MHC-II expression on HRSCs in patients with an interval of more than 12 months between ASCT and the PD-1-blocking therapy [26].

Similarly, the role of *9p24.1* genetic changes in predicting response to ICI was assessed in newly diagnosed cHL patients in Cohort D of the CheckMate 205 study. In this cohort, patients with advanced-stage cHL received four doses of nivolumab monotherapy followed by 12 doses of nivolumab plus AVD chemotherapy. HRSC *9p24.1* alterations (via FISH) were present in 22 baseline tumour biopsies, and a modified IHC H-score was used for the measurement of HRSC PD-L1 expression in 38 samples. All patients with evaluable *9p24.1* had detectable copy number alterations (CNAs), which included unbalanced rearrangement, amplification, copy gain, polysomy, and disomy. However, the types of CNAs were heterogenous between patient samples and within the same tumour biopsy. In this cohort, 55% (n = 12 patients) had an amplification of 9p24.1 as the overall highest-level CNA. Within each tumour biopsy, there were also disomic HRSCs, and the proportion of disomic HRSCs inversely correlated with the magnitude of *9p24.1* alterations (*p* = 0.01). The magnitude of *9p24.1* CNA was associated with a higher PD-L1 H-score although this finding was not significant (*p* = 0.067). There was a trend towards a higher nivolumab monotherapy response rates in patients with higher PD-L1 expression on HRSCs (*p* = 0.096). More patients with H-score PD-L1 expression in quartiles 3 or 4 compared to quartiles 1 or 2 had deeper and more durable responses to N-AVD (18/19 versus 12/19, *p* = 0.041) [89].

One interesting observation relates to the efficacy of PD1 inhibition patients with cHL who are refractory to anti-PD-L1 therapy. In a retrospective analysis of five patients with RR cHL who progressed on a phase II clinical trial for CS1001 (PD-L1 inhibitor), four demonstrated an excellent response when the treatment regimen transitioned to anti-PD1 therapy [90]. The median PFS for anti-PD-1 therapy was 27 months (range: 18 to 28 months). The amplification of *9p24.1* was observed in all five patients. The results of multiplex immunofluorescence showed the differential expression of PD-L1 and PD-L2 in cells in HRSCs and background cells including TAMs. PD-L1 and PD-L2 were colocalised on HRSC cells in 80% patients. These findings suggest that the efficacy of anti-PD1 therapy may be mediated not only through interactions with PD-L1 but also through PD-L2, which is not targeted in anti-PD-L1 therapy.

Ribas et al. showed that immune-related biomarkers, like the IFNγ signalling-related gene signature (analysed using the Nanostring nCounter as the average of the normalised expression levels of CXCL9, CXCL10, HLA-DRA, IDO-1, IFN-γ, and STAT1), have statistically significant associations with treatment response and PFS in patients with melanoma [91]. The IFN-γ-induced signature was analysed in 19 paired biopsy samples of RR-cHL patients (progressed on or after brentuximab vedotin) uniformly treated with pembrolizumab in the phase Ib study KEYNOTE-013 [92]. This gene signature was significantly upregulated after pembrolizumab (*p* = 0.017); however, it was not predictive of response in this small cohort.

To understand the significance of other biomarkers of ICI response, it is important to remember that antigen presentation to T-cells is a necessary condition for the effectiveness of these agents. In solid tumours, PD-1 efficacy is grounded in the activation of TME CD8+ T-cells via MHC-I molecules while in cHL, the alternate mechanism through MHC-II-mediated antigen presentation to CD4+ effector cells could be involved. This fact could be explained by frequent inactivating mutations and the copy loss of Beta-2-Microglobulin on HRSCs, which is necessary for the transportation of MHC-I molecules for antigen presentation to CD8+ T-cells [62]. Further evidence of the importance of CD4+ cells in the TME was elucidated by using multiplex immunofluorescence and DSP on the intact cHL TME, where PD-L1+ HRSCs were shown to be in direct physical contact with PD-1+/CD4+ T-cells more often than PD-1+/CD8+ T-cells [32]. Whilst several studies have shown the importance of CD4+ T-cell subsets, translational research in phase II ICI trials has yet to show their exact role in anti-tumour responses.

In the investigator-initiated phase II NIVAHL trial, 109 newly diagnosed patients with early-stage unfavourable cHL were randomised to either four cycles of nivolumab-AVD or sequential treatment with nivolumab monotherapy (four doses), then N-AVD, and then AVD (two cycles of each), followed by involved-site radiotherapy (30 Gray) in both groups. IHC PD-L1 expression was assessed in approximately 50 HRSCs in each case, and PD-L1 expression was categorised depending on the percentage of HRSCs with membranous staining (>50% of HRSCs was classified as ‘positive’, less than 50% as ‘less than positive’). MHC-I/II expression was categorised into the following groups: positive, with membranous staining in >50% of the tumour cells, and negative, with the staining in <50% of the tumour cells. The digital image analysis of all PD-L1+ cells (including HRSCs and the bystander cells) was performed. Similar to previous studies, there was a high variability in *9p24.1* CNAs observed in different tumour cells within the same patient. There was no impact of 9p24.1 CNAs on early response: the CR in 76% of patients was ‘copy gain’ versus the CR in 68% of patients, where it was ‘amplification’; *p* = 0.55. There was also no difference in CR rate between groups with different PD-L1 protein expression levels (72% CR in tumours with PD-L1 positivity versus 71% in tumours with ‘less than positive’ PD-L1 IHC; *p* = 1.00) and MHC-I expression (71% CR in positive MHC-I expression versus 74% in less-than-positive expression; *p* = 1.00) or MHC-II expression (77% CR in tumours positive for MHC-II versus 70% in less-than-positive expression; *p* = 0.60) in the whole cohort of patients. The analysis of the prognostic significance of *9p24.1* or PD-L1 expression was not feasible due to the excellent response and the one-year PFS rates in this trial of comparatively favourable patients with early-stage disease [93].

A recent focus of translational research has been to analyse immune changes in the TME and discover how these are reflected in the peripheral blood after ICI treatment. Reinke et al. analysed paired before- and on-treatment FFPE tissue biopsies to describe TME changes in ten patients treated in the NIVAHL study [94]. The study leveraged multiplex IHC and digital whole-slide image (WSI) analysis to describe both spatial changes within the TME as well as TCR sequencing and GEP (Nanostring) to evaluate changes in intra-tumoral and peripheral blood T-cell clonotypes. No single-cell RNA sequencing was performed for this analysis. Tumour biopsies and peripheral blood samples were taken during the first days after the commencement of treatment. All ten patients analysed in the NIVAHL cohort achieved either a partial or complete remission at the first interim assessment, limiting the ability to find biomarkers of ICI response in this study. The first key finding was the early clearance of HRSCs in the on-treatment biopsies of patients in the NIVAHL study. Remarkably, 50% (n = 5) of samples showed no HRSCs within biopsied lymphatic tissues after ICI therapy. Consistent with prior studies, MHC-I expression using IHC was absent in 67% (n = 6/9) of cases whilst MHC-II expression was absent in 20% of cases (two of ten). The second key finding was the absence of any cytotoxic T-cell anti-tumour response. Paired analysis showed no changes in the numbers of CD8+ T-cells in the on-treatment biopsies. TCR clonotypes also did not change after ICI in the peripheral blood or tumour, and there were no changes in the expression of T-cell cytotoxicity-related genes such as granzyme B upon GEP. Interestingly, while the CD8+ T-cell compartment remained static, there were changes in CD4+ T-cell subsets. The GEP of on-treatment biopsies showed the downregulation of genes associated with Type 1 Treg cells, including LAG3 and CTLA-4 but not PD-1. Type 1 Treg cells that express LAG3 were described by Aoki et al. as being associated with a trend towards inferior DFS [40]. Consistent with the GEP data, the level of PD-1 expression in the TME did not change after anti-PD1 therapy upon WSI analysis. The other key finding was that PD-L1 expression on TAMs that were spatially near HRSCs was strongly downregulated after ICI therapy. Taken together, these findings argue against a T-cell-mediated anti-tumour response. The authors hypothesised that reversed signalling through PD-L1 and the withdrawal of survival factors in HRSCs may be potential mechanisms of ICI response in cHL. These findings contrast with an earlier study by Cader et al. investigating peripheral blood immune changes in ICI-treated RR cHL patients [95]. This study showed an increased number of CD4+ TCR clonotypes and a distinct NK cell population, bearing the immunophenotype CD3−/CD68+/CD4+/GrB+, as being coincident with ICI response. However, the changes described by Cader et al. occurred later in the peripheral blood and so were not replicated by Reinke et al., in whose study only early immune changes were evaluated.

Garcia-Marquez et al. [96] subsequently evaluated immune-cell changes in peripheral blood from patients on the NIVAHL study. The authors used the flow cytometry of PBMCs at pre-specified timepoints after nivolumab therapy and correlated this with Nanostring gene expression in the tumour and Fluorospot assays looking at tumour antigen-specific T-cell responses. At baseline, patients had significantly fewer CD4+ T-cells compared to the healthy cohort. Both CD4+ and CD8+ T-cell populations had increased expression levels of co-inhibitory molecules, but the types of co-inhibitory molecules differed between these T-cell subsets. PD-1 overexpression was predominantly in CD8+ T-cells whereas poliovirus receptor (PVR) and LAG3 overexpression were in the CD4+ T-cell subset and CTLA4 and TIM3 were overexpressed in both. After nivolumab therapy, the level of co-inhibitory molecules, including PD-1, PVR, LAG3, TIM3, and CTLA4, decreased significantly in CD4+ and CD8+ T-cells after ICI, with the earliest response being seen after the first dose of nivolumab. TIGIT and CD96 levels, which were initially decreased at baseline, later increased after nivolumab therapy. However, the changes in co-inhibitory molecule expression on T-cells did not translate to any increase in tumour antigen-specific response. Despite these T-cell changes, there were no corresponding differences in before- and after-treatment responses to shared tumour antigens such as MAGE-A4 in terms of interferon gamma response. Patients with a higher interferon gamma response to one tumour-associated antigen at baseline were more likely to have an excellent response to nivolumab therapy (described as a >90% metabolic tumour volume reduction).

In summary, there are still no reliable prognostic biomarkers for ICI refractoriness, partly due to the excellent responses of patients with cHL to these agents. Currently, lower-level *9p24.1* alterations and low PD-L1 expression on HRSCs, as well as negative MHC-II expression on HRSCs in patients with a short interval between ASCT and the PD-1-blocking therapy, are associated with worse PFS in patients with RR cHL. Meanwhile, for newly diagnosed patients treated with ICI plus chemotherapy, there were no differences in the CR rate according to differing PD-L1 protein expression or tumour MHC-I or MHC-II expression in patients with early-stage disease; however, there was an association between deeper and more durable responses and higher PD-L1 IHC expression on HRSCs in patients with advanced-stage disease. Translational research has identified important changes in the TME after ICI therapy, and future research into these changes may mature into novel biomarkers that predict ICI response.

With growing evidence of new regimens for cHL yielding promising efficacy, particularly in the relapsed setting, incorporating various combinations of gemcitabine, vinblastine, liposomal doxorubicin, brentuximab, bendamustine, cyclophosphamide, etoposide, nivolumab, and pembrolizumab [15,97,98,99,100]; developing further knowledge of the immunogenicity of individual agents; and the conduction of translational studies to establish the interaction of the TME with these therapies are of upmost importance.

## 7. Conclusions

In this review, we have summarised the deeply complex subject of potential prognostic biomarkers within the TME in cHL and highlighted key immune-cell populations that have clinical significance with regard to response to therapy (refer to Table 1). HRSCs have genomic adaptations that result in cellular proliferation and immune evasion. Furthermore, HRSCs secrete a myriad of chemokines and cytokines that lead to the recruitment of an immunosuppressive TME.

Over the last decade, there has been a rapid development of new technologies to better understand the TME in cHL, including digital spatial profiling and single-cell RNA sequencing (refer to Table 2). This has provided vital information about not only the expression of genes in single cell resolution but also the spatial relationship between HRSCs and the diverse TME.

The key prognostic immune-cell populations highlighted in this review include cells of the mononuclear phagocyte system, including both TAMs and dendritic cells as well as distinct T-cell subpopulations. Not only the expression of immune checkpoint molecules on HRSCs but also the presence of immune cells within the TME, including LAG3, CTLA4, and PD-1/PD-L1, are not only prognostic biomarkers but also therapeutic targets. We are only beginning to understand the complexities of the interactions between HRSCs and the TME and between different immune cells within the TME. In addition, there is clear heterogeneity in the composition of the TME in cHL, further highlighting the need to better understand the TME and define prognostic features that are specific to the patient.

Future directions should also analyse the mechanisms of ICI efficacy in cHL to establish predictive biomarkers of response to ICI. More data using paired on-treatment biopsies at different time points, which are correlated with response data and immune changes in the peripheral blood, are required to better identify these predictive biomarkers. In addition, there is a need to translate the results of biomarker research into tests that are readily available to the clinician to provide the most effective treatment approaches for patients.

## Figures and Tables

**Figure 1 cancers-15-05217-f001:**
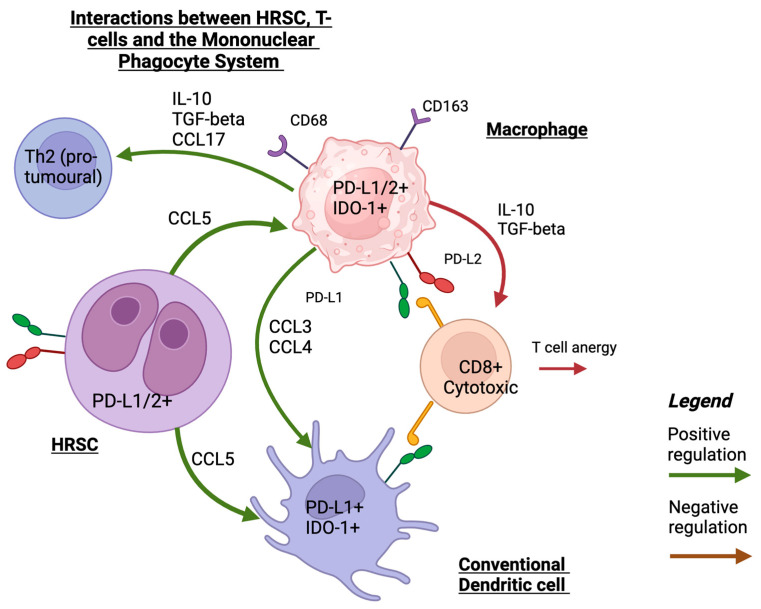
A schematic illustration of interaction between HRS, MPS, and T-cells that shows the main mechanisms of recruiting MPS cells into TME in HRSCs and interactions with T-cells. Created with Biorender.com (accessed on 24 September 2023).

**Figure 2 cancers-15-05217-f002:**
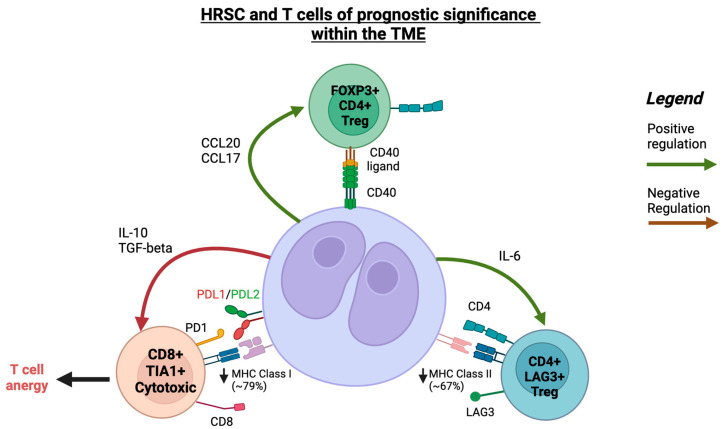
A schematic illustration of interaction between HRSCs and T-cells within the TME in cHL. Created with Biorender.com (accessed on 24 September 2023).

**Table 1 cancers-15-05217-t001:** Simplified overview of potential prognostic markers in cHL TME.

Proposed Biomarker	Clinical Setting	Impact on Outcome	Testing Method
Leukocytes			
High proportion of PD-L1+ leukocytes [65]	Treatment-naïve cHL	Inferior 5-year OS	Immunohistochemistry
TAMs and T-cells			
High checkpoint expression including PD-L1, TIM-3, LAG-3 and IDO-1 in TAMs and T-cell subsets [68]	Treatment-naïve cHL	Inferior 5-year OS	Gene expression profiling (Nanostring) and multiplex immunohistochemistry.
High levels of PD-1+ CXCL13+ T-follicular-like cells [40]	Treatment-naïve lymphocyte-rich cHL	Inferior 5-y PFS and Inferior 5-y OS	Multiplex immunofluorescence and single-cell RNA sequencing (10× genomics)
High proportion PD-1+, IDO-1+, LAG-3+ T-cells and macrophages [41]	RR cHL	Inferior 5-year OS	Gene expression profiling (Nanostring) and multiplex immunohistochemistry
HRSCs			
Low level of *9p24.1* alterations, lower H-score PD-L1+ expression, reduced MHC Class II expression in HRSCs [26]	RR cHL treated with ICI	Inferior PFS	Fluorescence in situ hybridisation and immunohistochemistry
Higher H-score PD-L1 expression in HRS [89]	RR cHL treated with ICI	Higher ORR	Fluorescence in situ hybridisation and immunohistochemistry

OS, overall survival; PFS, progression-free survival; HRSCs, Hodgkin Reed–Stenberg cells; TME, tumour microenvironment; ORR, objective response rate.

**Table 2 cancers-15-05217-t002:** A short description of the main principles of new laboratory techniques with pictorial illustrations and examples.

Laboratory Technique	Description of the Assay and Example Platform
Multiplex immunofluorescence [101]	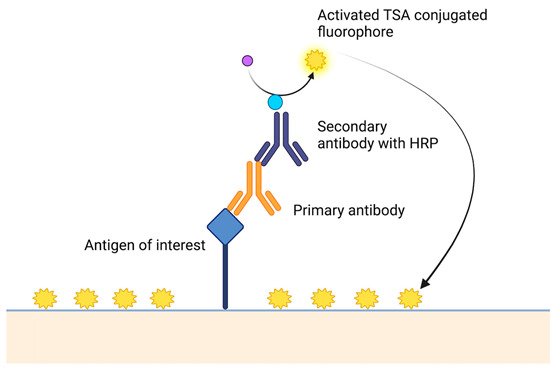 Vectra multispectral imaging platform using Opal chemistry (Akoya biosciences) This involves the immunostaining of a tissue using a primary antibody to bind the antigen, which is followed by a secondary antibody conjugated to horseradish peroxide. This activates a TSA-conjugated fluorophore. Each antigen has a unique fluorophore that emits light at non-overlapping wavelengths. The TSA-conjugated fluorophore covalently links to the tissue, amplifies the fluorescent signal, and improves the signal-to-noise ratio. After the immunostaining, there is sequential image acquisition using a fluorescent microscope. This enables the detection of up to eight targets in a single image acquisition stage.
Digital spatial profiling & spatial transcriptomics [102]	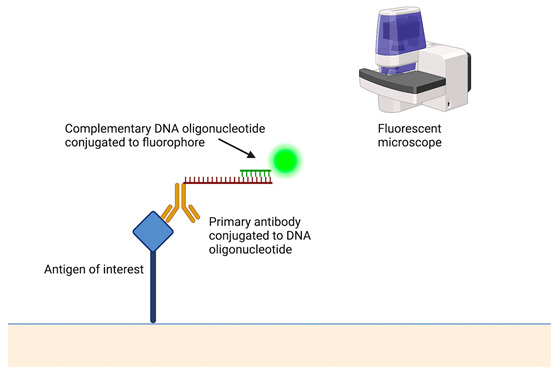 Akoya Phenocycler (Akoya biosciences) This enables the visualisation of the gene or protein expression at a cellular level. It utilises antibodies conjugated to a specific DNA oligonucleotide that binds to the antigen of interest. A second complementary DNA oligonucleotide with a fluorophore then hybridises to the primary antibody. Using an automated microfluidics system and a fluorescent microscope, the sequential hybridisation, imaging, and stripping of the fluorescently labelled probes occur. This allows the visualisation of up to 60 markers in a single tissue section with the benefits of minimizing spectral overlap and batch effects.
Single-cell RNA sequencing [103]	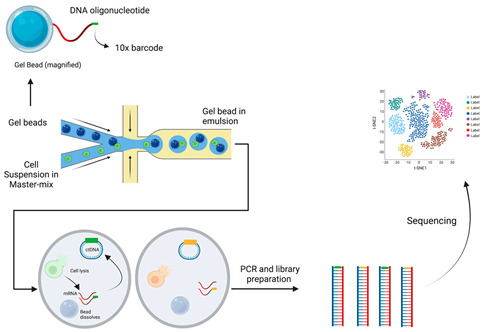 10× Genomics Single-cell RNA sequencing enables the analysis of differences in gene expression within distinct immune-cell populations in the tumour microenvironment. This provides different information from bulk RNA sequencing, which provides an average transcript level overall to describe all cells within the TME. Firstly, a cell suspension is partitioned into individual cells using a GEM proprietary microfluidics system. Each GEM contains either a single cell or no cell and a bead. The bead is attached to a barcoded oligonucleotide fragment that has a 10× barcode, which is unique to each GEM/bead. There is also a UMI that allows normalisation for differences in PCR amplification between transcripts. After GEM creation, the bead dissolves and the cell is lysed, releasing the oligonucleotide fragment to bind to the target mRNA. This mRNA is converted to cDNA using the reverse transcriptase process (reverse transcriptase is part of a master mix that is combined with the cell suspension prior to GEM creation), and the cDNA is amplified in a PCR reaction to create multiple libraries from each cell. The libraries are then sequenced, and the presence of the 10× barcode, which is unique to each cell in the assay, enables the analysis of gene expression in single cell resolution.
Gene expression profiling [104]	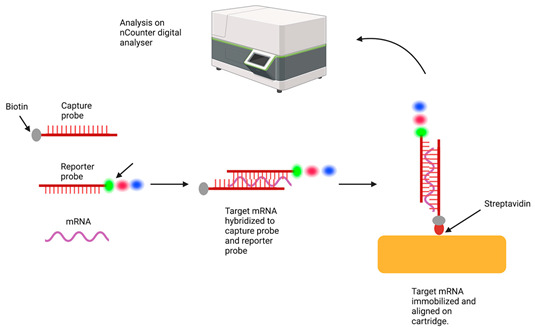 Nanostring nCounter Technology This is used to perform both differential gene expression analysis as well as the cellular deconvolution of the gene expression data that describe the proportions of immune cells in the TME. RNA is hybridised with a biotin-labelled capture probe and a reporter probe containing a fluorescent molecular barcode. Following this, the target RNA is immobilised on a cartridge coated with streptavidin, which conjugates with the biotin-labelled capture probe. The samples are purified by removing unbound RNA and excess probes, then aligned on the automated nCounter Prep station. After this, the samples are transferred to the nCounter Digital analyser, which reads the molecular barcodes. The quantitative counts are proportional to the levels of gene expression. Normalisation to the levels of expression of reference genes allows comparisons between different samples and different batches.

TSA, tyramide signal amplification; DNA, deoxyribonucleic acid; RNA, ribonucleic acid; GEM, gel-in-bead emulsion; PCR, polymerase chain reaction; UMI, unique molecular identifier; cDNA, copy deoxyribonucleic acid; figures created with Biorender.com (accessed on 24 September 2023).

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
