# Peer review of "Prognostic Markers within the Tumour Microenvironment in Classical Hodgkin Lymphoma"

_cancers, 2023, doi:10.3390/cancers15215217_

Round 1

Reviewer 1 Report

Comments and Suggestions for Authors

The authors should give more emphasis on clinical impact of drugs targeting tumor microenvironment, in particular tumor-associated macrophages (TAM) and distint T-cell sub-populations. In this respect, the authors should add in the text and in the bibliography two recent reports. The first report regard the clinical efficacy of a liposomal doxorubicin super-charge containing regimen ( Picardi M et al. Br J Haematol 2022; 198: 847-860) and the role of non-pegylated liposomal doxorubicin on TAM. The second report regards the clinical efficacy of a combined treatment with brentuximab vedotin and bendamustine (Picardi M, et al. Blood Advance 2019; 14: 1546-1552) and their role on bystander effect, immunogenic cell death and antibody-dependent cellular phagocytosis.

Comments on the Quality of English Language

Minor editing

Author Response

Thank you for reviewing our manuscript and for the recommendation. The manuscript focuses on the prognostic role of the TME in HL, rather than the efficacy of individual regimens specifically. As suggested, we have now included the aforementioned references and some other therapies in the following paragraph to highlight suggested papers and others we felt relevant to the "Prognostic immune checkpoint biomarkers in the TME" section:

With growing evidence of new regimens for HL yielding promising efficacy, particularly in the relapsed setting, incorporating various combinations of gemcitabine, vinblastine, liposomal doxorubicin, brentuximab, bendamustine, cyclophosphamide, etoposide, nivolumab and pembrolizumab [15,97-100], further knowledge of the immunogenicity of individual agents, and translational studies to establish the interaction of the TME with these therapies are of upmost importance.

 With the following additional references cited:

  1. Moskowitz, A.J.; Shah, G.; Schöder, H.; Ganesan, N.; Drill, E.; Hancock, H.; Davey, T.; Perez, L.; Ryu, S.; Sohail, S.; et al. Phase II Trial of Pembrolizumab Plus Gemcitabine, Vinorelbine, and Liposomal Doxorubicin as Second-Line Therapy for Relapsed or Refractory Classical Hodgkin Lymphoma. Journal of Clinical Oncology2021, 39, 3109-3117, doi:10.1200/jco.21.01056.
  2. Mei, M.G.; Lee, H.J.; Palmer, J.M.; Chen, R.; Tsai, N.C.; Chen, L.; McBride, K.; Smith, D.L.; Melgar, I.; Song, J.Y.; et al. Response-adapted anti-PD-1-based salvage therapy for Hodgkin lymphoma with nivolumab alone or in combination with ICE. Blood 2022, 139, 3605-3616, doi:10.1182/blood.2022015423.
  3. Advani, R.H.; Moskowitz, A.J.; Bartlett, N.L.; Vose, J.M.; Ramchandren, R.; Feldman, T.A.; LaCasce, A.S.; Christian, B.A.; Ansell, S.M.; Moskowitz, C.H.; et al. Brentuximab vedotin in combination with nivolumab in relapsed or refractory Hodgkin lymphoma: 3-year study results. Blood 2021, 138, 427-438, doi:10.1182/blood.2020009178.
  4. Picardi, M.; Della Pepa, R.; Giordano, C.; Pugliese, N.; Mortaruolo, C.; Trastulli, F.; Rascato, M.G.; Cappuccio, I.; Raimondo, M.; Memoli, M.; et al. Brentuximab vedotin followed by bendamustine supercharge for refractory or relapsed Hodgkin lymphoma. Blood Adv 2019, 3, 1546-1552, doi:10.1182/bloodadvances.2019000123.
  5. Picardi, M.; Giordano, C.; Pugliese, N.; Esposito, M.; Fatigati, M.; Muriano, F.; Rascato, M.G.; Della Pepa, R.; D'Ambrosio, A.; Vigliar, E.; et al. Liposomal doxorubicin supercharge-containing front-line treatment in patients with advanced-stage diffuse large B-cell lymphoma or classical Hodgkin lymphoma: Preliminary results of a single-centre phase II study. Br J Haematol 2022, 198, 847-860, doi:10.1111/bjh.18348.

Reviewer 2 Report

Comments and Suggestions for Authors

The authors review tumor immunity in Hodgkin lymphoma from a variety of perspectives, including many discussions of antigen presentation to T cells and Tregs, which will be useful to readers of this journal.
The Tregs are divided into effector Tregs and exhausted Tregs, and these fractions are also well reviewed in solid tumors. I recommend discussing the literature on these cells in Hodgkin lymphoma.

Author Response

We thank you for the highlighting these important T-cell sub populations. We now have added a paragraph to the T-cell section to include this in the manuscript in the "T-cell subsets of prognostic significance in cHL TME" section:

The role of CD4+ Treg cells with an exhausted phenotype has been evaluated in number of malignancies [83]. Recent work by Veldman et al. investigated the characteristics of CD4+ T-cells surrounding HRSC which are known to lack CD26 expression (a marker of T-cell activation) [84,85].  Using the results of bulk RNA sequencing validated with single cell RNA sequencing and immunohistochemistry, Veldman et al. showed that CD4+CD26- T-cells are located in close proximity to HRSC and have a predominantly antigen experienced Treg memory phenotype. In addition, the CD4+CD26- T-cell population has increased gene expression of thymocyte selection-associated high mobility group box (TOX) and TOX2 transcription factors. These are known to be upregulated in T-cell states with chronic activation and also result in increased expression of PD-1 and CXCL13.  However, the prognostic impact of CD4+ CD26- T-cells and expression of TOX/TOX2 and the role of Treg cells in modifying ICI response in cHL remains to be better elucidated.

We have also included the following relevant additional references:

  1. Nagasaki, J.; Togashi, Y. A variety of 'exhausted' T cells in the tumor microenvironment. Int Immunol 2022, 34, 563-570, doi:10.1093/intimm/dxac013.
  2. Veldman, J.; Rodrigues Plaça, J.; Chong, L.; Terpstra, M.M.; Mastik, M.; van Kempen, L.C.; Kok, K.; Aoki, T.; Steidl, C.; van den Berg, A.; et al. CD4+ T cells in classical Hodgkin lymphoma express exhaustion associated transcription factors TOX and TOX2: Characterizing CD4+ T cells in Hodgkin lymphoma. Oncoimmunology 2022, 11, 2033433, doi:10.1080/2162402x.2022.2033433.
  3. Ma, Y.; Visser, L.; Blokzijl, T.; Harms, G.; Atayar, C.; Poppema, S.; van den Berg, A. The CD4+CD26- T-cell population in classical Hodgkin's lymphoma displays a distinctive regulatory T-cell profile. Lab Invest 2008, 88, 482-490, doi:10.1038/labinvest.2008.24.